# Protocol for a systematic review and meta-analysis of the diagnostic test accuracy of host and HPV DNA methylation in cervical cancer screening and management

Sarah J Bowden,[1,2] Laura Burney Ellis [ORCID] ,[1,2] Ilkka Kalliala,[2,3] Maria Paraskevaidi,[2] Jack Tighe,[2] Konstantinos S Kechagias,[2] Triada Doulgeraki,[4] Evangelos Paraskevaidis [ORCID] ,[5] Marc Arbyn,[6] James Flanagan [ORCID] ,[1] Areti Veroniki,[7] Maria Kyrgiou[1,2]

SJB and LBE are joint first authors.

For numbered affiliations see end of article.

**Correspondence to**
Sarah J Bowden;
s.bowden@imperial.ac.uk

## ABSTRACT

**Introduction** Human papillomavirus (HPV) is necessary but not sufficient for cervical cancer development. During cervical carcinogenesis, methylation levels increase across host and HPV DNA. DNA methylation has been proposed as a test to diagnose cervical intraepithelial neoplasia (CIN); we present a protocol to evaluate the accuracy of methylation markers to detect high-grade CIN and cervical cancer.

**Methods and analysis** We will search electronic databases (Medline, Embase and Cochrane Library), from inception, to identify studies examining DNA methylation as a diagnostic marker for CIN or cervical cancer, in a cervical screening population. The primary outcome will be to assess the diagnostic test accuracy of host and HPV DNA methylation for high-grade CIN; the secondary outcomes will be to examine the accuracy of different methylation cut-off thresholds, and accuracy in high-risk HPV positive women. Our reference standard will be histology. We will perform meta-analyses using Cochrane guidelines for diagnostic test accuracy. We will use the number of true positives, false negatives, true negatives and false positives from individual studies. We will use the bivariate mixed effect model to estimate sensitivity and specificity with 95% CIs; we will employ different bivariate models to estimate sensitivity and specificity at different thresholds if sufficient data per threshold. For insufficient data, the hierarchical summary receiver operating curve model will be used to calculate a summary curve across thresholds. If there is interstudy and intrastudy variation in thresholds, we will use a linear mixed effects model to calculate the optimum threshold. If few studies are available, we will simplify models by assuming no correlation between sensitivity and specificity and perform univariate, random-effects meta-analysis. We will assess the quality of studies using QUADAS-2 and QUADAS-C.

**Ethics and dissemination** Ethical approval is not required. Results will be disseminated to academic beneficiaries, medical practitioners, patients and the public.

**PROSPERO registration number** CRD42022299760.

## STRENGTHS AND LIMITATIONS OF THIS STUDY

⇒ The results of this meta-analysis will guide targeted further research into specific genes as diagnostic markers for cervical precancer and cancer.
⇒ We will separately analyse diagnostic test accuracy in high-risk human papillomavirus positive women, in addition to a general screening population, in order to determine the suitability of DNA methylation as a triage tool as well as a diagnostic test.
⇒ A test with higher diagnostic test accuracy for cervical intraepithelial neoplasia than cytology would have clear clinically significant impact for use as a triage or screening tool.
⇒ Our results will be limited by the quality and quantity of studies in the review.
⇒ Variation in measuring and reporting methylation may lead us to conduct several separate meta-analyses, rather than pooling all data.

## INTRODUCTION
### Rationale

Cervical cancer continues to be the fourth most common cancer in women globally, resulting in 311 000 deaths in 2018.[1 2] Human papillomavirus (HPV) is responsible for 99% cervical cancer; about 14 genotypes of HPV are carcinogenic and classified as high-risk HPV types (hrHPV).[3 4] Over 70% of women are infected with HPV during their lifetime.[5] In a small number, persistent hrHPV leads to development of precancerous lesions, which can progress to cancer.[6]

Cervical precancer (cervical intraepithelial neoplasia (CIN), classified into three histological grades CIN1-3), can usually be treated surgically. Where implemented, call-recall cervical screening programmes have dramatically reduced incidence of and mortality from

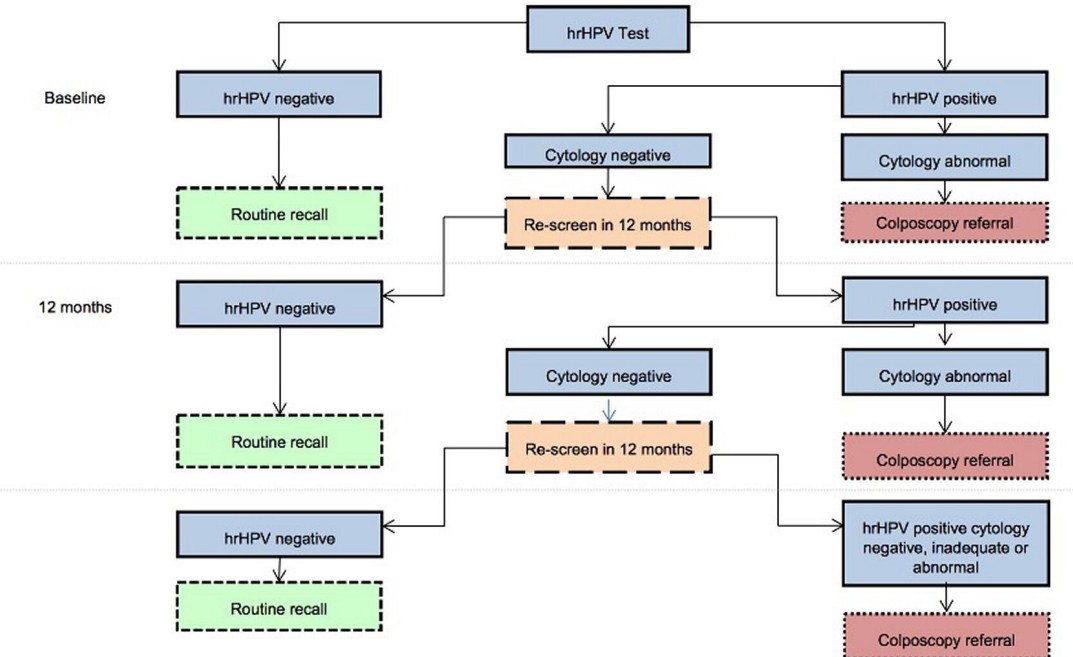

**Figure 1** NHSCSP HPV primary screening programme. This highlights the role of the triage test (currently cytology within UK) in the cervical screening programme algorithm, for management of women detected as hrHPV positive at primary screening, and onward decision tree for colposcopy referral vs repeat screening in 12 months. HPV, human papillomavirus; hrHPV, high-risk HPV; NHSCSP, NHS Cervical Screening Programme.

cervical cancer through monitoring and treatment of CIN.[2 7] Such programmes have traditionally used cytology of exfoliated cervical cells as the primary screening test. Cytology testing has limitations as a screening test including a low sensitivity for CIN (53%–75%),[8–10] high interobserver and intraobserver variation and minimal future potential for robotic automation, point-of-care testing and self-sampling. As a result, much research into alternative tests to cytology has been undertaken in recent years.

HrHPV DNA has been shown to be more sensitive than cytology for detection of precancerous lesions and cancer,[11] and is now the primary screening method of choice in many countries, including the UK.[12] Primary hrHPV DNA screening is anticipated to offer 60%–70% greater protection against invasive cancer compared with cytology-based screening.[13] However, higher sensitivity for CIN is associated with a decreased specificity, resulting in a modest positive predictive value,[14] in part due to the fact that the hrHPV positive prevalence in countries such as the UK is as high as 13%, reaching up to 28% in women aged 30 years or less.[15] A triage test for those found to be positive for hrHPV is needed to improve the specificity of screening and identify only those who are truly at risk of precancerous and cancerous lesions.

Reflex cytology (cytology testing for hrHPV-positive samples on the same specimen) remains the test with the best historic safety data and is part of the national guidelines in many countries, including the UK[16] (figure 1). However, molecular tests are of increasing interest, particularly human (host) and hrHPV methylation, HPV

E6 and E7 mRNA testing and overexpression of proteins indicating progression (precancer). Such tests may offer the potential to improve accuracy in diagnosis of progressive CIN, while also allowing machine automation, self-sampling and point-of-care testing in the future.

Methylation is a regulatory chemical process, which mostly occurs at specific methylation sites in DNA, called methylation 5-cytosine-phosphate-guanine-3 (CpG) sites. This causes the switching 'on and off' of specific genes to affect gene expression and determine downstream cellular processes. Methylation has been noted to be an early and frequent event in the development of many cancers, and is commonly observed at high levels in tumour tissues.[17]

Studies of DNA methylation so far have focused on the variation in levels of methylation in different grades of CIN and cancer, in both human and HPV genes. Aberrant methylation levels have been noted during carcinogenesis; typically global hypomethylation, with an increase in methylation levels in tumour suppressor or tumour suppressor-like genes.[18] An increase in methylation in particular genes has been reported to be associated with increasing grades of cervical disease.[19–21] Diagnostic accuracy studies of human and HPV methylation as markers of high-grade cervical neoplasia have shown promising results,[22 23] with good sensitivity and specificity.[20 24] A meta-analysis of HPV genes found HPV L1/L2 pooled sensitivity to be 77%,[20] and a systematic review of methylation in host DNA found methylation of FAM19A4 to have a sensitivity between 68.2%–86.7% and specificity 60.6%–79.3%.[21] Studies have reported a range of thresholds for

cut-off values for DNA methylation as a diagnostic test and examined methylation.

Human and/or hrHPV methylation tests may be able to fulfil the urgent unmet need for a triage test for HPV positive women diagnosed at screening. This systematic review and meta-analysis aims to explore the diagnostic test accuracy of human and/or HPV methylation tests; in women of cervical screening age, whether a positive methylation test can accurately diagnose High-grade Squamous Intraepithelial Lesion (HSIL)/CIN2+, as defined by histology as the reference standard. Our aim is to identify the best performing candidate gene markers.

## Objectives
### Primary objective
The primary objective is to determine the diagnostic test accuracy of human and viral DNA methylation tests for the detection of CIN grade 2 or worse (CIN2/HSIL+), as defined by histology as a reference standard, in women of cervical screening age.

### Secondary objectives
A secondary objective is to evaluate the diagnostic accuracy of human and HPV DNA methylation tests for the detection of CIN2/HSIL+, in different target populations for the use in cervical screening (where studies are available), specifically:
► Primary screening.
► Triage of cytology positive women.
► Triage of hrHPV positive women.
► Post treatment of CIN.

Further objectives include evaluation of the diagnostic accuracy of human and HPV DNA methylation for detection of CIN3/HSIL+, or invasive cervical cancer (ICC).

We also aim to explore the effect of different cut-off thresholds that are provided for single (or combined) methylation CpG sites (human and/or HPV) in diagnosing CIN2+ or CIN3+.

Also, in the presence of any comparator test in the same study (eg, cytology or HPV16/18 genotyping), we will analyse relative accuracy of DNA methylation to the alternative test. Finally, in the event of sufficient data comparing self-collected and clinician-collected samples, we will analyse the relative accuracy of both methods of collection.

## METHODS AND ANALYSIS
This protocol was written in accordance with Preferred Reporting Items for Systematic Review and Meta-Analysis Protocols guidelines.[25]

## Patient and public involvement
A patient and public involvement (PPI) meeting was conducted prior to the start of planning this meta-analysis. We discussed the current evidence surrounding methylation testing with patients and cervical cancer charity leaders, and the implications this might have on future cervical cancer screening. We discussed the need for prioritisation of early detection, while avoiding over treatment and anxiety related to false positive (FP) diagnoses. The research question was defined with the aim of assessing the accuracy of methylation markers, as this was highlighted by our PPI meeting.

## Eligibility criteria
### Types of participants
Any population of adult women representing a cervical screening setting or referral population will be included, without demographic restrictions, including retrospective biobanks of samples from women recruited from such populations, regardless of hrHPV genotype and sample material. Where data are available, we will separately analyse hrHPV positive women.

### Index tests to be considered
All DNA methylation tests will be considered: human DNA methylation tests, hrHPV DNA methylation tests or tests which combine CpG sites from human and hrHPV tests into a single marker.

Although there are several million potential CpG sites in the human genome, we will not limit the review to any particular genes and will report all findings. Within the HPV genome there are eight genes, containing several hundred CpG sites, which vary according to HPV genotype. All genes will be considered but will be analysed separately according to HPV type. Assays can only be developed for a single genomic region spanning 2–300 base-pairs however many tests include multiple sites by combining results of different genes into one test score. We will not restrict to either single or multiple site methylation tests but will aim to analyse these separately where possible.

All human DNA and hrHPV DNA methylation technologies will be included, for example, we will include (but not limited to) bisulphite clonal sequencing, pyrosequencing, epiTYPER, next-generation sequencing (NGS), Luminex, methylation-sensitive high resolution melting, methylation-specific PCR (MSP) and Illumina 450k and 850k arrays.[26] Both in-house laboratory assays and commercial assays will be included. As many technologies are still under investigation, all technologies and cut-off thresholds will be considered.

### Target condition
The primary target condition is high-grade cervical preinvasive disease (CIN2 or CIN3) or cervical cancer. This will be measured as CIN 2 or worse (CIN2+) for squamous cell lesions and adenocarcinoma in situ glandular lesions, as defined by histological analysis of a cervical biopsy. Where low-grade squamous intraepithelial lesion and HSIL terminology is used to define CIN identified by histology, a HSIL or worse (HSIL+) definition will be accepted as equivalent to CIN2 or worse (CIN2/HSIL+) (table 1). For some studies a threshold of CIN3 or worse

**Table 1** Summary of cytology and histology classification systems in use by NHSCSP for squamous and glandular lesions

| Cytology | Histology | |
|---|---|---|
| NHSCSP/BAC 2021 | Bethesda 2014 | NHSCSP 2012 |
| Borderline changes in squamous cells | ASC-US ASC-H | |
| Low-grade dyskaryosis | LSIL | CIN1 |
| High-grade dyskaryosis (moderate) | HSIL | CIN2 |
| High-grade dyskaryosis (severe) | HSIL | CIN3 |
| High-grade dyskaryosis/? invasive SCC | HSIL SCC | SCC |
| Borderline changes in endocervical cells | AGC NOS | |
| ? Glandular neoplasia of endocervical type | Endocervical | L-CGIN |
| ? Glandular neoplasia (non-cervical) | Endometrial | |
| | Glandular | |
| | AGC favours neoplastic | H-CGIN |
| | Endocervical | |
| | Glandular | |
| | Endocervical AIS | |
| | Adenocarcinoma | ACC |
| | Endocervical | |
| | Endometrial | |
| | Extrauterine | |
| | NOS | |

ACC, adenocarcinoma; AGC, atypical glandular cells; AIS, adenocarcinoma in situ; ASC-H, atypical squamous cells cannot exclude HSIL; BAC, British association for cytopathology; CIN, cervical intraepithelial neoplasia; H-CGIN, high-grade cervical glandular intraepithelial neoplasia; HSIL, high-grade squamous intra-epithelial lesion; L-CGIN, low-grade-CGIN; LSIL, low-grade squamous intraepithelial lesion; NHSCSP, NHS Cervical Screening Programme; NOS, not otherwise specified; SCC, squamous cell carcinoma.

(CIN3+) by histology may have been used. These will be included but analysed separately.

## Reference standard

Histology, performed on a cervical tissue biopsy taken at colposcopy examination, is the preferred reference standard for diagnosis of high-grade CIN (CIN2 or worse) or cervical cancer and will be included as according to colposcopy guidelines in the most recent NHS Cervical Screening Programme (NHSCSP) guidelines.[16 27] Where the colposcopic examination is abnormal, we will assume that the histological material taken as a reference standard is fully representative of the disease status. Of note,

the NHSCSP 2020 guidelines state that women with a negative hrHPV test do not need to be referred to colposcopy for biopsy. Additionally, a tissue biopsy is not clinically indicated in the presence of a normal colposcopy. Therefore, for ethical reasons, we will not require a histological biopsy for the normal grading of disease where there is either a negative hrHPV test, normal cytology or a normal colposcopy.

## Types of studies

Randomised control trials, cross-sectional studies, longitudinal cohort studies, and both prospective and retrospective case–control studies, will be included where they assess the diagnostic test accuracy of human or HPV methylation marker to detect high-grade preinvasive cervical disease as confirmed by histology. We will include studies assessing the accuracy of methylation markers for diagnosis only; we will not include studies assessing methylation markers as predictive markers for future development of cervical cancer or preinvasive cervical disease.

Case reports will be excluded, as will studies in cell lines or tumour clones only. Studies that used only cytology as a reference standard will be excluded, and studies which do not use colposcopy directed histological biopsy for confirmation of a positive disease status (CIN1 or worse). We will exclude any studies where the number of participants with CIN2+ is less than five which are likely to give unreliable overall estimates.

## Information sources and search strategy

The literature search will start from inception (1946) and will include studies published up to the present day (26 April 2023).

Separate search strategies will be developed for MEDLINE Ovid and Embase Ovid.[28] Searches will be conducted without language or publication status restrictions. A draft Medline strategy can be found in online supplemental appendix 1. For databases other than MEDLINE, the search strategy will be adapted accordingly. The following trial registries will be searched for ongoing studies: US National Institutes of Health Ongoing Trials Register ClinicalTrials.gov; WHO International Clinical Trials Registry Platform.

We will also search the MetaRegister of Controlled Trials (mRCT), National Cancer Institute (NCI) Physicians Data Query (PDQ), ClinicalTrials.gov and NCI Clinical Trials will be searched for ongoing studies. The main investigators of ongoing studies will be contacted for further information in case of unclear data, to determine study eligibility. Forward citation searching will be performed in Scopus/Web of Science. To identify studies which might have been missed during the electronic search, the citation lists of included studies and relevant reviews will be hand-searched and experts in the field, including directors of UK cancer and colposcopy registries, contacted to identify further reports of studies. We will further check the proceedings of conferences via the Conference Proceeding Citation Index-science, and

unpublished eprints via MedRxiv and specifically the following relevant conferences:

► Annual Meeting of the British Society of Colposcopy and Cervical Pathology.
► Annual Meeting of the International Federation of Cervical Pathology and Colposcopy.
► EUropean Research Organisation on Genital Infection and Neoplasia.
► European Federation of Colposcopy Congresses.
► Annual Meeting of the American Society of Colposcopy and Cervical Pathology.

Both published and unpublished studies will be included if they meet the inclusion criteria for the systematic review.

The review is registered with PROSPERO; the registration number is CRD42022299760. The registered study end date is 31 December 2023.

### Selection process

We will use reference management database EndNote.[29] Two review authors will screen all titles and abstracts. We will exclude those studies which clearly do not meet the inclusion criteria and will obtain copies of the full text of potentially relevant references. Two review authors will independently assess the eligibility of retrieved papers. We will then compare the results and resolve any disagreements by discussion. Disagreements will be resolved by consensus, with discussion with a third review author if necessary. We will document reasons for exclusion.

### Data extraction

Two review authors will independently perform data extraction using a data extraction form. Disagreements will be resolved by consensus, with discussion with a third review author if necessary. We will extract data on the following items:

► Study design, population and setting: author, year, study design, any randomisation, study population, geography, ethnicity, total number of participants, hrHPV status (positive or negative), hrHPV assay used to define positivity, hrHPV genotype of HPV positive participants.
► Index test: technology, CpG site(s) and location, gene, hrHPV genotype for HPV tests, cut-off threshold used.
► Sample material, for example, vaginal swab, cervical swab, endocervical curettage, urine.
► Results: true positives (TP), FPs, true negatives (TNs), false negatives will be extracted for each test and outcome (including for comparator test where available).
► Comparator: any comparator test including details of the technology.
► Reference standard: histological grade, cytological grade.
► Outcome measures, for example, CIN2+, CIN3+, ICC; (where individual patient-level outcome data are not available, where possible, we will contact authors to obtain this).

We will extract data for different HPV types separately and for both individual CpG sites and combinations of CpG sites where available. We will use hg38, and where studies have used hg19 or hs1, we will use LiftOver to convert these to hg38, using a LiftOver package in R.[30] Where data are missing (ie, individual CpG sites data, or the absolute number of true-positive or false-positive or negative), we will contact the authors and collate additional data for studies with missing data. Where data for individual CpG sites are not provided by the author, we will perform the analysis at the gene level, or using combinations of CpG sites and present this appropriately. Our priority outcome is CIN2+, as this is currently the most clinically significant diagnosis.[31]

### Risk of bias in individual studies

Two authors will assess the risk of bias for eligible studies using the Quality Assessment of Studies of Diagnostic Accuracy included in Systematic reviews (QUADAS)-2 tool independently.[32 33] We will assign a judgement of low, moderate or high risk of bias to each of the domains (A) patient selection, (B) index test, (C) reference test and (D) flow and timing, according to the criteria outlined in the Cochrane Handbook.[34] We will assess each domain using the signalling questions and answering yes (Y), no (N) or unclear (U). We will tailor the signalling questions to the review as summarised in online supplemental appendix 1. An overall judgement will be formed for risk of bias by recording information used to support the risk of bias judgement, the signalling questions and the judgement of high, low or unclear. For applicability, we will record information on the applicability of each study, and rate as low, high or unclear (no score is applied for the flow and timing domain under applicability as according to QUADAS-2 guidance). For comparator studies, we will use the QUADAS-C tool to assess risk of bias.[35]

We will check multiple sources to decrease the possibility of reporting and publication bias (for example, searching grey literature, citations of included studies and relevant reviews). We will assess whether only small studies (those with less than 100 participants) provide supporting evidence and will conduct a sensitivity analysis to assess robustness of results using large studies only (studies with 100 participants or more).

### Data synthesis

In order to define methylation positivity, we will aim to explore and summarise different cut-off points of methylation used in the literature. When several studies present duplicate data from the same population, the study with the largest sample size will be selected for inclusion in the analysis.

For absolute diagnostic accuracy, we will use the TP, TN, false negative (FN) and FP, as defined by the individual studies in terms of presence of high-grade CIN or cancer (CIN2/HSIL+) in the index test against the reference standard. Where data for the absolute sensitivity and

specificity for the detection of CIN3+ or cervical cancer is provided, we will calculate this separately.

It is important to note that different HPV genotypes and genes require different test assays, and these cannot be included in the same meta-analyses. We will, therefore, aim to pool accuracy from tests that identify similar targets, which we anticipate will involve several different analyses:

1. Analysis of tests using the same HPV genotype and CpG sites (human or viral).
2. Analysis of tests using the same HPV genotypes and genes (individual CpG sites are not available).
3. Analysis of tests using multiple CpG sites across different genes and viral or human genomes (where the exact same combination of multiple CpG sites has been performed in different populations).

Different technologies (such as QMSP and pyrosequencing) will be combined in the above analyses, if the same CpG sites or same genes have been included.

We will first restrict to studies reporting a common threshold and will use the bivariate model[36] to estimate a summary sensitivity and specificity. Each study will contribute to the analysis once using TP, FN, TN and FP (or 2×2 table). (Where multiple studies report the same cut-off threshold in addition to other thresholds, we will choose to include data corresponding to the most commonly reported cut-off threshold. For example, we will include sensitivity and specificity from two studies evaluating methylation of the same gene, where both studies report a 10% methylation threshold.)

If there are sufficient data, we will additionally explore how different thresholds impact on the estimation of sensitivity and specificity using separate bivariate models. If there are no sufficient data to perform separate bivariate models, we will apply the hierarchical summary receiver operating curve model[37] to estimate a summary curve from studies that have used different test thresholds. In the case of multiple thresholds within each study, with different thresholds between studies, we will perform a linear mixed effects model accounting for heterogeneity and correlation between groups and using the TP, TN, FP and FN values from all thresholds of each study. This model will estimate the optimal threshold, that is, the point where the test is likely to perform best using the maximum value of the weighted sum of sensitivity and specificity.[38] Analyses will be performed using the glmer function in the R package lme4[39] and the metadata Stata command.[40] When the 2×2 data are incomplete across studies and cannot be reconstructed (eg, sensitivity and/or specificity estimates are available only), we will simplify models by assuming no correlation between sensitivity and specificity and perform two univariate, random-effects, standard meta-analyses for sensitivity and specificity separately. We will use the metaprop function in the R package meta.[41] If few studies are available in a meta-analysis (eg, fewer than three), and effect estimates are similar enough to be combined in a meta-analysis, then a fixed effect meta-analysis will be performed, since

heterogeneity variances are expected to be poorly estimated and it is inappropriate to overfit models by estimating too many parameters from few studies.[42]

In the case of any studies presenting concomitant data for methylation tests compared with alternative tests (cytology or HPV16/18 genotyping), we will examine relative accuracy. In R, for meta-analysis of diagnostic test accuracy studies, the pooled relative sensitivity and specificity of the index tests versus comparator tests will be assessed by including the test as a covariate in the bivariate model.[34 43] We will compute ratios of sensitivities and specificities across tests. Finally, where possible, we will compute pooled positive predictive values and negative predictive values, and transform this into a pre–post test probability plot, to facilitate the understanding of how these results can impact clinical practice.

## Sensitivity analyses

We plan to complete sensitivity analyses by restricting to studies free of concern for the QUADAS-2 domains based on the signalling questions, we have identified as the most important. Therefore, sensitivity analyses will be performed including only those studies answering yes for each of the four domains (1–4), where:

1. Appropriate patient selection where selection was consecutive or random (QUADAS-2 item 1).
2. Where the index test used an appropriate methylation test and sample material (QUADAS-2 items 6 and 7).
3. Appropriate reference standard was used (QUADAS-2 item 8).
4. Flow and timing was appropriate (QUADAS-2 items 10 and 12).

We will perform an additional sensitivity analysis where case–control studies are excluded, and a further analysis where all types of studies except randomised control trials are excluded, in order to view their effect on the results.

By default, studies using older technologies for the methylation test will be excluded under the QUADAS-2 domain 2; in this way, we will perform a sensitivity analysis for studies where only an appropriate methylation test was used. Although it will not be possible to evaluate the accuracy of technologies themselves in this review, we will use prior expert knowledge from the literature on the most accurate technologies to inform the sensitivity analyses. In this way, we will explore the effect of restricting to high-quality technologies, such as pyrosequencing and NGS on diagnostic test accuracy estimates.

## Additional analyses

We will assess the between-study heterogeneity through visual inspection of forest plots (variability in sensitivity and specificity). We will also sort paired forest plots by sensitivity to explore if a reverse trend in estimates of specificities is detected. If there is variation in threshold, we will also visually inspect whether the study-specific effects lie close to the summary ROC curve. The higher the scatter of the study-specific effect sizes and the larger the prediction ellipse, the higher the between-study heterogeneity.

We will explore if variability can be explained by sample size, threshold effects, test variations, study characteristics or potential effect modifiers. If there are more than 10 studies available in a meta-analysis, we will explore heterogeneity by adding covariate terms using meta-regression.

We intend to assess the influence of the following characteristics in a subgroup or meta-regression analysis: the geographical location where the study was conducted (continent; high vs low and middle income country), age groups (<30, 30–59, 60+), premenopausal and postmenopausal populations, the ethnicity of the study population, the technology of test used, complete versus partial verification with the reference standard used, the type of index test used (viral, human or mixed methylation; targeted genes), clinician sample or self-sample), and commercial versus non-commercial tests.

## Quality of evidence

We will perform Grading of Recommendations, Assessment, Development and Evaluations (GRADE)[44] to assess our confidence in the results and our recommendations.

## What this study adds

Previous meta-analyses of methylation markers as diagnostic tests have been published.[24] While our study provides an update of over 4 years of data to previous studies, we also aim to address previous methodological issues with an exploration of both the generalised population of women of screening age and the specific population of HPV positive women detected at screening, with heterogeneity analyses, and sensitivity analyses to separately analyse single CpG sites. We will use a linear mixed effects model which will allow us to present the optimum threshold of both sensitivity and specificity. We will display our results in forest plots, and employ a pre–post test probability plot to facilitate direct potential clinical interpretation of data for cervical screening programmes. We will assess certain characteristics, including age, in subgroup meta-regression analyses. Importantly, we will perform a validated risk of bias assessment (QUADAS-2 and QUADAS-C) and perform a sensitivity analysis excluding studies with a high risk of bias. We will also quantify the certainty of the evidence using GRADE.

## ETHICS AND DISSEMINATION

This review does not require ethical approval.

We identified four groups of potential stakeholders (academic beneficiaries; health-related agencies and decision-makers; medical practitioners; patients and public) and specific action items to effectively target them. We will publish our review in an open access journal and will present its findings at international conferences (eg, European Research Organisation on Genital Infection and Neoplasia, International Papillomavirus Conference and European Society of Gynaecological Oncology Congress). We will make the datasets available to the wider research community. We will organise a workshop with key stakeholders. We will develop information sheets and briefings, highlighting the key findings and circulate newsletters. We will engage the press with presentations and social media interviews and we will work closely with Jo's Trust charity that plays an important role in educating patient communities.

**Author affiliations**
[1]Department of Surgery & Cancer, Faculty of Medicine, Institute of Reproductive and Developmental Biology, Imperial College, London, UK
[2]Department of Metabolism, Digestion and Reproduction, Faculty of Medicine, Institute of Reproductive and Developmental Biology, Imperial College, London, UK
[3]Department of Obstetrics and Gynaecology, University of Helsinki and Helsinki University Hospital, Helsinki, Finland
[4]Department of Obstetrics & Gynaecology, Ioannina University Hospital, Ioannina, Greece
[5]Department of Obstetrics & Gynaecology, University of Ioannina, Ioannina, Greece
[6]Unit of Cancer Epidemiology, Belgian Cancer Centre, Brussels, Belgium
[7]Institute of Health Policy Management and Evaluation, St Michael's Hospital Toronto, Toronto, UK

**Contributors** SJB, IK and MK conceived of the study. SJB, LBE and IK designed the Medline search strategy and will perform the search. SJB and MK designed the study objectives and inclusion/exclusion criteria. SJB, MA and MK conceived of the approach to the QUADAS-2 tool for methodological quality assessment. SJB, AV and IK have developed the statistical methodology and analysis tools. All authors (SJB, LBE, AV, IK, MP, KSK, JT, EP, MA, JF and MK) have been involved in the drafting of the protocol. MK is the guarantor of the review.

**Funding** SB received funding from Wellcome Trust 4i Clinician Scientist Fellowship (P77712) and LE received funding from Imperial Health Charity (RFPR2223_27).

**Competing interests** None declared.

**Patient and public involvement** Patients and/or the public were involved in the design, or conduct, or reporting, or dissemination plans of this research. Refer to the Methods section for further details.

**Patient consent for publication** Not applicable.

**Provenance and peer review** Not commissioned; externally peer reviewed.

**Author note** AV and MK are joint last authors.

**ORCID iDs**
Laura Burney Ellis http://orcid.org/0000-0002-0639-785X
Evangelos Paraskevaidis http://orcid.org/0000-0002-1859-261X
James Flanagan http://orcid.org/0000-0003-4955-1383

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
