## [Reviewer comments · BMJ Open]

ARTICLE DETAILS

TITLE (PROVISIONAL)	Protocol for a Systematic Review and Meta-analysis of the Diagnostic Test Accuracy of host and HPV DNA methylation in cervical cancer screening and management
AUTHORS	Bowden, Sarah J; Ellis, Laura; Kalliala, Ilkka; Paraskevaïdi, Maria; Tighe, Jack; Kechagias, Konstantinos S.; Doulgeraki, Triada; Paraskevaïdis, Evangelos; Arbyn, Marc; Flanagan, James; Veroniki, Areti; Kyrgiou, Maria

VERSION 1 – REVIEW

REVIEWER	Herzog, Chiara University of Innsbruck, European Translational Oncology Prevention and Screening Institute
REVIEW RETURNED	27-Feb-2023

GENERAL COMMENTS	The manuscript by Bowden*, Ellis* et al. describes the protocol for a systematic review and meta-analysis of recent studies on host and human papillomavirus (HPV) DNA methylation diagnostic accuracy, especially sensitivity and specificity, in cervical cancer screening. Triage of high-risk HPV positive individuals is an important aspect in cervical cancer prevention and novel strategies that outperform cytology as a current strategy for colposcopy referral are urgently required. The manuscript is generally well-written, and the work proposed by Bowden, Ellis et al. follows good practice standards by relying on established bias assessment and reporting guidelines. Among several promising candidate triage (and or/screening) tests based on DNA methylation of viral or host genes, the systematic study of the potential diagnostic accuracy is highly relevant and the results are expected to be interesting to the identified stakeholders, including academic community, medical practitioners, and potential patients/the public. Some aspects of the manuscript require further clarification and/or may need to be amended to answer study questions. These points are outlined below and relate primarily to how results can best be presented/interpreted across the current landscape of (unfortunately) heterogeneous studies on methylation and CIN, and the uniqueness of the current study in the context of prior systematic reviews and meta-analyses. Major aspects: 1. The current study references previous systematic reviews and meta-analyses with highly similar aims and premises as the current study, in particular a previous systematic review and meta-analysis Kelly et al. 2019 (https://doi.org/10.1038/s41416-019-0593-4). While the current study is more recent by at least 3 years and can hence add more studies and data, its unique and original
---

aspects compared to prior reports, as well as how new value is added, should be further emphasized and clarified in the protocol (for instance: broader criteria of studies and populations).

2. The analysis aims to evaluate methylation overall, regardless of modality; will results be presented separately for different modalities and/or target genes, highlighting potential bias in underlying studies? Can the authors comment how an overall comparison will best be presented? This is an important aspect as it relates to whether the study design can address the question asked by this review.

3. While the risk of bias will be evaluated using QUADAS-2 and QUADAS-c, and the authors state that studies with less than 5 CIN2+ cases excluded, it is unclear how they will deal with imbalances in analysis that depend on the underlying population and cannot be accounted for in higher-level statistics (true/false positive/negative cases).
Although more prospective studies are emerging, many studies still depend on (nested) case-control studies and measures such as sensitivity and specificity can be influenced by the number and characteristics of cases (and controls). As an example: the sensitivity for most DNA methylation-based assays is higher for CIN3 than CIN2; hence, if 99% of "CIN2+" patients are actually CIN3 and only 1% is CIN2, the sensitivity is expected to be higher than if 99% are CIN2 and only 1% is CIN3 (although these are of course extreme examples). Similarly, the underlying age distribution is expected to influence performance of assays (a majority of assays have primarily been tested in women above 30 only and generally perform worse in women below 30). This heterogeneity could possibly be investigated if additionally looking at the level of 'individual' patient-level data rather than summary statistics of the overall respective studies - given such data are provided by the original respective study authors – and could be highly interesting/informative to the academic readership. Could the authors clarify whether such aspects will be considered and addressed?

4. Diagnostic accuracy measures: sensitivity and specificity are important, but in a clinical setting positive and negative predictive values are perhaps equally if not more relevant and facilitate decision-making for practitioners or policy makers. Hence it could be informative to evaluate and/or compare these additional metrics in this review study, accounting for the variety of settings (depending on prevalence, which is different in general setting versus hrHPV, etc.). For instance, the PPV estimate could inform how many individuals in a given population need to be referred to further assessments (e.g. colposcopy +/- biopsy) for a 'true positive' diagnosis.

5. Threshold selection: since the authors are including a range of technologies and genes (both host and HPV methylation), can the authors further clarify how they will combine/identify optimal "thresholds" across modalities? While the methodology is in general clear, thresholds are likely specific per methodology/gene. Would such a step only apply if several studies utilise the same technology and gene?

Minor aspects:

1. Methylation levels (Abstract, Page 3 Line 20 and Introduction, Page 5 Line 43): while it is true that many assays rely on increased methylation levels, often due to simplicity of the corresponding hypermethylation assay, the blanket statement that during cervical carcinogenesis methylation levels increase in host DNA is not

	strictly true as also hypomethylation can be observed during progression to cervical carcinogenesis (see e.g. https://doi.org/10.1186/s13073-022-01116-9); it might be more straightforward to state that “aberrant methylation occurs” (although hypermethylation is most frequently exploited for specific assays). 2. Introduction, Page 5 Line 13, it is not clear that reference 14 provides information on the numbers of hrHPV positivity in the UK. 3. Search Strategy, Page 9 Line 39, Please add exact dates for study search period (inception and present day). 4. DNA methylation tests show great potential, in part because they have the possibility to utilise self-collected samples and could therefore encourage individuals that do not attend current screening modalities to participate in cervical cancer screening. Should this be considered when including studies that e.g. utilise endocervical curettage methods, and/or should these be considered separately?
--	---

REVIEWER	Ladoukakis , Efthymios Queen Mary University of London, Wolfson Institute of Population Health
REVIEW RETURNED	12-Mar-2023

GENERAL COMMENTS	 - Protocol papers should report either planned or ongoing studies. The PROSPERO registration the authors provide states as the study start date the 31st of January 2022 and study end date the 31st of December 2022. Is this a planned study? If yes the start and end dates need to be updated and included in the manuscript. - In Page 5 / line 6: Reference [12] is mostly about the highest sensitivity of hrHPV testing than cytology. It does not mention hrHPV testing being the primary screening choice (yet) of any countries including UK. - In Page 5 / line 31: Methylation mostly occurs at CpG sites. It can occur in other sites as well. - In Page 6 / line 23: It is unclear what “in different target populations” means. -In Page 7 / line 21: I suggest you mention the Illumina arrays (e.g. EPIC) as a lot of your methylation data will come from these technologies. - In page 8 / line 22: Why is there a question mark (?) in glandular neoplasia? - In page 9 / lines 20-25: If you are going to include both prospective and retrospective case-control studies then you will have two different types of tests; the one would include early stage predictions (if the disease will appear in the next "x" months/years) and the other would include diagnostic tests (if the disease is present). The objectives and planning of your analysis should take into consideration this differentiation. -In Page 10 / “Data extraction”: there is no mention about including the version of genome assembly (e.g. hg38) for the CpG locations and the strategy to convert everything to a common assembly (e.g. LiftOver)
---

	- In Page 11 / lines 49-51: "We will assess... large studies only". I think this sentence needs clarification. Do you mean you will check whether only small studies can be useful or only large studies can be useful? - In Page 17 / lines 12-13: There seem to be a problem with the references' format - In Page 19 / Appendix 1: The search strategy needs some clarification. It seems that some lines are mixed up.
--	---

VERSION 1 – AUTHOR RESPONSE

Reviewer: 1

Dr. Chiara Herzog, University of Innsbruck

Comments to the Author:

The manuscript by Bowden*, Ellis* et al. describes the protocol for a systematic review and meta-analysis of recent studies on host and human papillomavirus (HPV) DNA methylation diagnostic accuracy, especially sensitivity and specificity, in cervical cancer screening. Triage of high-risk HPV positive individuals is an important aspect in cervical cancer prevention and novel strategies that outperform cytology as a current strategy for colposcopy referral are urgently required. The manuscript is generally well-written, and the work proposed by Bowden, Ellis et al. follows good practice standards by relying on established bias assessment and reporting guidelines. Among several promising candidate triage (and or/screening) tests based on DNA methylation of viral or host genes, the systematic study of the potential diagnostic accuracy is highly relevant and the results are expected to be interesting to the identified stakeholders, including academic community, medical practitioners, and potential patients/the public.

Some aspects of the manuscript require further clarification and/or may need to be amended to answer study questions. These points are outlined below and relate primarily to how results can best be presented/interpreted across the current landscape of (unfortunately) heterogeneous studies on methylation and CIN, and the uniqueness of the current study in the context of prior systematic reviews and meta-analyses.

Thank you for your positive feedback on our protocol. We appreciate your comments and the time taken to review our work.

Major aspects:

1. The current study references previous systematic reviews and meta-analyses with highly similar aims and premises as the current study, in particular a previous systematic review and meta-analysis Kelly et al. 2019

([https://gbr01.safelinks.protection.outlook.com/?url=https%3A%2F%2Fdoi.org%2F10.1038%2Fs41416-019-0593-](https://gbr01.safelinks.protection.outlook.com/?url=https%3A%2F%2Fdoi.org%2F10.1038%2Fs41416-019-0593-4&data=05%7C01%7C1.ellis7%40nhs.net%7C6f7ec9f954274c3d8d0008db2f98d0a1%7C37c354b285b047f5b22207b48d774ee3%7C0%7C0%7C638156105600590308%7CUnknown%7CTWFpbGZsb3d8eyJWIjojMC4wLjAwMDAiLCJQIjoiV2luMzliLCJBTiI6Ikk1haWwiLCJXVCI6Mn0%3D%7C3000%7C%7C%7C&sdata=plc1%2F1CZDodbaDXbRDBpXA3RCLkFmldRs81uz6BCOUc%3D&reserved=0)

[4&data=05%7C01%7C1.ellis7%40nhs.net%7C6f7ec9f954274c3d8d0008db2f98d0a1%7C37c354b285b047f5b22207b48d774ee3%7C0%7C0%7C638156105600590308%7CUnknown%7CTWFpbGZsb3d8eyJWIjojMC4wLjAwMDAiLCJQIjoiV2luMzliLCJBTiI6Ikk1haWwiLCJXVCI6Mn0%3D%7C3000%7C%7C%7C&sdata=plc1%2F1CZDodbaDXbRDBpXA3RCLkFmldRs81uz6BCOUc%3D&reserved=0](https://gbr01.safelinks.protection.outlook.com/?url=https%3A%2F%2Fdoi.org%2F10.1038%2Fs41416-019-0593-4&data=05%7C01%7C1.ellis7%40nhs.net%7C6f7ec9f954274c3d8d0008db2f98d0a1%7C37c354b285b047f5b22207b48d774ee3%7C0%7C0%7C638156105600590308%7CUnknown%7CTWFpbGZsb3d8eyJWIjojMC4wLjAwMDAiLCJQIjoiV2luMzliLCJBTiI6Ikk1haWwiLCJXVCI6Mn0%3D%7C3000%7C%7C%7C&sdata=plc1%2F1CZDodbaDXbRDBpXA3RCLkFmldRs81uz6BCOUc%3D&reserved=0)).

While the current study is more recent by at least 3 years and can hence add more studies and data, its unique and original aspects compared to prior reports, as well as how new value is added, should be further emphasized and clarified in the protocol (for instance: broader criteria of studies and populations).

Thank you for highlighting this. The number of studies in this field is increasing year on year with increasing interest in DNA methylation as a clinical triage test. There has been a move from more experimental studies to diagnostic test accuracy studies in cervical screening populations, which are more generalisable, since 2018. The review by Kelly et al, 2019 BJC provides a good summary and analysis of data on the potential of DNA methylation tests as diagnostic markers of cervical lesions. However, there are several limitations in that review that we plan to overcome with our approach. Firstly, this meta-analysis pooled studies with highly variable cut-off thresholds with no exploration of the impact of differing cut-offs on the sensitivity and specificity of tests. Furthermore, they used a standardised specificity (70% or 50%) to compare tests rather than standardising by cut-off thresholds (which balances sensitivity and specificity), making interpretation for test performance more difficult. Perhaps most importantly, there was insufficient description of methodological quality assessment for studies, and no sensitivity analyses based on quality or risk of bias. Heterogeneity between studies is not mentioned or addressed and forest plots were not provided for visual inspection of heterogeneity. Furthermore, the effect of differing methylation assays on test results is not addressed or accounted for. Another major omission is that differences in CpG sites in tests are not described in the main paper or supplement, meaning it is not possible to know if the same or different CpG sites are being compared between studies.

By developing and publishing this protocol we intend to outline a clear and robust methodological approach which overcomes the issues in our study, in order to provide an excellent up to date meta-analysis of methylation for both Host and HPV gene markers in CIN and cervical cancer that is clearly interpretable for those working in the field.

Whilst these points are written in our protocol methodology, we have added a section 'what this study adds' (page 14, line 12) in order to succinctly highlight the intention of this review, which is as follows:

'What this study adds

Previous meta-analyses of methylation markers as a diagnostic test have been published [25]. Whilst our study provides an update of over 4 years of data to previous studies, we also aim to address previous methodological issues with an exploration of both the generalised population of women of screening age and the specific population of HPV positive women detected at screening, with heterogeneity analyses, and sensitivity analyses to separately analyse single CpG sites. We will use a linear mixed effects model which will allow us to present the optimum threshold of both sensitivity and specificity. We will display our results in forest plots, and employ a pre-test post-test probability plot to facilitate direct potential clinical interpretation of data for cervical screening programmes. We will assess certain characteristics, including age, in sub-group meta-regression analyses. Importantly, we will perform a validated risk of bias assessment (QUADAS-2 and QUADAS-C), and perform a sensitivity analysis excluding studies with a high risk of bias. We will also quantify the certainty of the evidence using GRADE.'

2. The analysis aims to evaluate methylation overall, regardless of modality; will results be presented separately for different modalities and/or target genes, highlighting potential bias in underlying studies? Can the authors comment how an overall comparison will best be presented? This is an important aspect as it relates to whether the study design can address the question asked by this review.

Our primary aim is to identify the best performing tests, as measured by diagnostic test accuracy, dependent on Host or HPV (or a combination) genes. All analyses will be presented separately by gene unless a combined gene test has been used, where this will also be analysed as a combination test where multiple studies exist. A secondary aim is to explore the effect of different methylation technologies on diagnostic accuracy and heterogeneity.

We are aware that some technologies are less accurate in measuring methylation which as you point out, can introduce bias; this may particularly arise with older studies using older methods before pyrosequencing and next generation sequencing were introduced. We will address this through our QUADAS-2 assessment and performing a sensitivity analysis on studies with a low ROB, which excludes those where the index test may have introduced bias due to its known inaccuracy, as described in the QUADAS-2 methods. Although it will not be possible to evaluate the accuracy of technologies themselves in this review, we will use prior expert knowledge from the literature on the most accurate technologies to inform the sensitivity analyses. By performing sensitivity analyses restricting to studies using pyrosequencing only, for example, we can explore the heterogeneity that studies using less robust methods may have introduced. In this way we will explore the effect of restricting to high-quality technologies on diagnostic test accuracy estimates.

We have added a sentence to the end of the introduction (page 4, line 3):

'This systematic review and meta-analysis aims to explore the diagnostic test accuracy of human and/or HPV methylation tests; in women of cervical screening age, whether a positive methylation test can accurately diagnose High-grade Squamous Intraepithelial Lesion (HSIL)/CIN2+, as defined by histology as the reference standard. Our aim is to identify the best performing candidate gene markers.'

We have also added a paragraph in the sensitivity analysis section (page 13, line 16):

'By default, studies using older technologies for the methylation test will be excluded under the QUADAS-2 domain b; in this way we will perform a sensitivity analysis for studies where only an appropriate methylation test was used. Although it will not be possible to evaluate the accuracy of technologies themselves in this review, we will use prior expert knowledge from the literature on the most accurate technologies to inform the sensitivity analyses. In this way we will explore the effect of restricting to high-quality technologies, such as pyrosequencing and next generation sequencing on diagnostic test accuracy estimates.'

3. While the risk of bias will be evaluated using QUADAS-2 and QUADAS-c, and the authors state that studies with less than 5 CIN2+ cases excluded, it is unclear how they will deal with imbalances in analysis that depend on the underlying population and cannot be accounted for in higher-level statistics (true/false positive/negative cases).

Although more prospective studies are emerging, many studies still depend on (nested) case-control studies and measures such as sensitivity and specificity can be influenced by the number and characteristics of cases (and controls). As an example: the sensitivity for most DNA methylation-based assays is higher for CIN3 than CIN2; hence, if 99% of "CIN2+" patients are actually CIN3 and only 1% is CIN2, the sensitivity is expected to be higher than if 99% are CIN2 and only 1% is CIN3 (although these are of course extreme examples). Similarly, the underlying age distribution is expected to influence performance of assays (a majority of assays have primarily been tested in women above 30 only and generally perform worse in women below 30). This heterogeneity could possibly be investigated if additionally looking at the level of 'individual' patient-level data rather than summary statistics of the overall respective studies - given such data are provided by the original respective study authors – and could be highly interesting/informative to the academic readership. Could the authors clarify whether such aspects will be considered and addressed?

Thank you for this important comment. We would like to address these two points.

Firstly, regarding the number of cases and controls in specific populations, we agree, the data will be skewed where there are a higher number of patients with a 'worse' diagnosis in a given population. To

overcome heterogeneity within populations, where possible, we will use CIN2+, CIN3+ and ICC as separate endpoints to understand the difference in performance. Where this is not possible, we contact authors of the original studies in order to request individual patient level data.

To clarify this, we have added cervical cancer as an end-point in our outcome measures as follows (page 10, line 20)

‘• Outcome measures e.g. CIN2+, CIN3+, Invasive Cervical Cancer (ICC); (where individual patient-level outcome data is not available, where possible, we will contact authors to obtain this)’

Secondly, thank you for your comment, regarding age. We agree, this is a very important aspect, as methylation data has been shown to vary depending on the age of the population. As stated in our additional analyses paragraph (page 13, line 23), we will perform a subgroup meta-regression analysis based on age (we currently plan to do groups <30, 30-59, and 60+). We believe this will be valuable information to provide to readers, and we aim to present this data in similar groups to cervical screening age group populations (for example, in many countries screening begins around age 30, and is performed less frequently in those over 60).

4. Diagnostic accuracy measures: sensitivity and specificity are important, but in a clinical setting positive and negative predictive values are perhaps equally if not more relevant and facilitate decision-making for practitioners or policy makers. Hence it could be informative to evaluate and/or compare these additional metrics in this review study, accounting for the variety of settings (depending on prevalence, which is different in general setting versus hrHPV, etc.). For instance, the PPV estimate could inform how many individuals in a given population need to be referred to further assessments (e.g. colposcopy +/- biopsy) for a ‘true positive’ diagnosis.

Thank you. As most diagnostic studies tend to report sensitivity and specificity, or Area Under the Curve, we anticipate that the majority of data will be provided in this format and plan to analyse these metrics for the primary outcome. However, we agree that positive predictive value is more clinically relevant, particularly in a screening context. We will therefore add PPV and NPV as outcomes for both general screening and hrHPV positive populations and analyse these separately where data are available.

For this reason we plan to additionally analyse both negative predictive values and positive predictive values. Based on a presumed prevalence of CIN2 and CIN3 in a high-risk HPV positive population, we will apply these values and display them in a pre-test post-test probability plot, which will allow the reader to understand the potential context in which a methylation marker could allow triage of patients into or out of colposcopy.

We have added this into our protocol (page 12, line 33):

‘Finally, where possible, we will compute pooled positive predictive values and negative predictive values, and transform this into a pre-test post-test probability plot, to facilitate the understanding of how these results can impact clinical practice.’

5. Threshold selection: since the authors are including a range of technologies and genes (both host and HPV methylation), can the authors further clarify how they will combine/identify optimal “thresholds” across modalities? While the methodology is in general clear, thresholds are likely specific per methodology/gene. Would such a step only apply if several studies utilise the same technology and gene?

Thank you. We agree that thresholds are likely specific per gene and even CpG site, whilst comparison of different technologies risks introducing bias. From preliminary searches, we have found that studies may report a range of cut-off thresholds for the same gene. Where possible, we will compare to the same cut-off positivity threshold e.g. 10% methylation threshold.. We recognise that

there is likely to be significant heterogeneity in the technologies used, and therefore this will be analysed in a sensitivity analysis where we will restrict to certain technologies considered as gold-standard and explore the risk of bias and heterogeneity. To make this clearer, we have added a sentence to the data synthesis section as follows (page 12, line 33):

'We will first restrict to studies reporting a common threshold and will use the bivariate model [36] to estimate a summary sensitivity and specificity. Each study will contribute to the analysis once using TP, FN, TN, and FP (or 2x2 table). (Where multiple studies report the same cut-off threshold in addition to other thresholds, we will choose to include data corresponding to the most commonly reported cut-off threshold. For example, we will include sensitivity and specificity from two studies evaluating methylation of the same gene, where both studies report a 10% methylation threshold.)'

Minor aspects:

1. Methylation levels (Abstract, Page 3 Line 20 and Introduction, Page 5 Line 43): while it is true that many assays rely on increased methylation levels, often due to simplicity of the corresponding hypermethylation assay, the blanket statement that during cervical carcinogenesis methylation levels increase in host DNA is not strictly true as also hypomethylation can be observed during progression to cervical carcinogenesis (see e.g.

Thank you, this is important to clarify in our manuscript. We have amended the manuscript as follows, and added the reference Saghafinia et al, Cell Reports 2018 (page 4, line 23):

'Studies of DNA methylation so far have focused on the variation in levels of methylation in different grades of CIN and cancer, in both human and HPV genes. Aberrant methylation levels have been noted during carcinogenesis; typically global hypomethylation, with an increase in methylation levels in tumour suppressor or tumour suppressor-like genes [19]. An increase in methylation in particular genes has been reported to be associated with increasing grades of cervical disease'

2. Introduction, Page 5 Line 13, it is not clear that reference 14 provides information on the numbers of hrHPV positivity in the UK.

Thank you for noticing this. This reference [previously 14, now 15] was in support of the first clause of the sentence. We have now updated this, and added a second reference (Rebolj et al, BMJ 2019 [16]). We have updated the text in line with this reference as follows (page 4, line 5):

'However, higher sensitivity for CIN is associated with a decreased specificity, resulting in a modest positive predictive value [15], in part due to the fact that the hrHPV positive prevalence in countries such as the UK is as high as 13%, reaching up to 28% in women aged 30 years or less [16].'

3. Search Strategy, Page 9 Line 39, Please add exact dates for study search period (inception and present day).

Thank you. We kept 'present day' open to allow for an updated search once we are ready to run the analyses. However, for the purpose of this protocol we have added specific dates, which may need to change dependent on acceptance of publication (page 9, line 2):

'The literature search will start from inception (1946) and will include studies published up to the present day (26th April 2023).'

4. DNA methylation tests show great potential, in part because they have the possibility to utilise self-collected samples and could therefore encourage individuals that do not attend current screening modalities to participate in cervical cancer screening. Should this be considered when including studies that e.g. utilise endocervical curettage methods, and/or should these be considered separately?

Thank you for this important comment. Firstly, this will be evaluated in our Risk of Bias assessment QUADAS-2, and we will perform a sensitivity analysis including studies where the sample is taken from the cervix alone, such as endocervical curettage (page 13, line 9).

We agree that a significant advantage of methylation tests are that they can be performed on self-collected samples, and this is one of the important reasons that we require an updated assessment of their accuracy. We believe there is benefit in analysing and presenting the accuracy of DNA methylation as diagnostic markers, regardless of the collection method. HPV testing has been shown to have a similar but slightly lower accuracy on self-samples as compared to clinician-collected (Arbyn et al, Lancet Oncol 2014), and initial data comparing methylation tests has been shown to be similar (De Strooper et al, Gynaecol Oncol, 2016).

Whilst our primary focus is the performance of methylation as a diagnostic test, if there are sufficient data to separately analyse methylation markers on self-collected samples, this would be a useful addition to our study. We have therefore added the following to the manuscript (page 5, line 23): 'Also, in the presence of any comparator test in the same study (e.g. cytology or HPV16/18 genotyping), we will analyse relative accuracy of DNA methylation to the alternative test. Finally, in the event of sufficient data comparing self-collected and clinician-collected samples, we will analyse the relative accuracy of both methods of collection.'

Reviewer 2:

Dr. Efthymios Ladoukakis , Queen Mary University of London

Comments to the Author:

1. Protocol papers should report either planned or ongoing studies. The PROSPERO registration the authors provide states as the study start date the 31st of January 2022 and study end date the 31st of December 2022. Is this a planned study? If yes the start and end dates need to be updated and included in the manuscript.

Thank you for your feedback and your comments.

Thank you for this observance. This is a planned study. We have updated the study end date on PROSPERO to 31st December 2023, and we have included this in the manuscript as follows (page 9, line 27):

'The review is registered with PROSPERO; the registration number is CRD42022299760. The registered study end date is 31st December 2023.'

2. In Page 5 / line 6: Reference [12] is mostly about the highest sensitivity of hrHPV testing than cytology. It does not mention hrHPV testing being the primary screening choice (yet) of any countries including UK.

Thank you for noticing this. Reference 12 is affiliated to the first clause of the sentence regarding the sensitivity of high-risk HPV testing as compared to cytology. We have moved reference 12 and added reference 13 (NHSCSP Cervical Screening: Programme Overview. Available at:

<https://www.gov.uk/guidance/cervical-screening-programme-overview>).

3. In Page 5 / line 31: Methylation mostly occurs at CpG sites. It can occur in other sites as well.

Thank you, this is an important detail which we have now amended in the manuscript as follows (page 4, line 17):

'DNA methylation is a regulatory chemical process, which mostly occurs at specific sites in DNA, where cytosine and guanine nucleotides are adjacent called methylation 5-cytosine-phosphate-guanine-3 (CpG) sites'

4. In Page 6 / line 23: It is unclear what "in different target populations" means.

Thank you for your comment. We were referring to the populations listed in bullet form below. We have amended the manuscript to try and make this more clear, as follows (page 5, line 12):

'A secondary objective is to evaluate the diagnostic accuracy of human and HPV DNA methylation tests for the detection of CIN2/HSIL+, in different target populations for the use in cervical screening (where studies are available), specifically:

- Primary screening
- Triage of cytology positive women
- Triage of high-risk HPV positive women
- Post treatment of CIN'

5. In Page 7 / line 21: I suggest you mention the Illumina arrays (e.g. EPIC) as a lot of your methylation data will come from these technologies.

Thank you for this comment. We have amended the manuscript accordingly as follows (page 6, line 27):

'All human DNA and hrHPV DNA methylation technologies will be included: for example we will include (but not limited to) bisulphite clonal sequencing, pyrosequencing, epiTYPER, next generation sequencing (NGS), Luminex, Methylation-Sensitive High Resolution Melting (MS-HRM), Methylation-Specific PCR (MSP), and Illumina 450k and 850k arrays [24].'

6. In page 8 / line 22: Why is there a question mark (?) in glandular neoplasia?

In the NHSCSP/BAC (British Association for Cytology) guidelines (updated in 2021) [<https://www.gov.uk/government/publications/cervical-screening-laboratory-hpv-testing-and-cytology-services/cervical-screening-guidance-for-laboratories-providing-hpv-testing-and-cytology-services-in-the-nhs-cervical-screening-programme>], Glandular Neoplasia is denoted in official terms with a question mark preceding it. On reviewing this, we note that the guidelines have been updated in 2021 from 2013, and we have updated our manuscript accordingly: now two terms have a preceding question mark as per the guidelines. We have now referenced the updated guideline (2021, in place of 2013) (page 7, line 11):

Cytology Histology

NHSCSP/BAC 2021 Bethesda 2014 NHSCSP 2012

Borderline changes in squamous cells ASC-US

ASC-H

Low-grade dyskaryosis LSIL CIN1

High-grade dyskaryosis (moderate) HSIL CIN2

High-grade dyskaryosis (severe) HSIL CIN3

High-grade dyskaryosis/?invasive SCC HSIL

SCC SCC
Borderline changes in endocervical cells AGC NOS
?Glandular neoplasia of endocervical type Endocervical L-CGIN
?Glandular neoplasia (non-cervical) Endometrial
Glandular
AGC favours neoplastic H-CGIN
Endocervical
Glandular
Endocervical AIS
Adenocarcinoma ACC
Endocervical
Endometrial
Extrauterine
NOS

7. In page 9 / lines 20-25: If you are going to include both prospective and retrospective case-control studies then you will have two different types of tests; the one would include early stage predictions (if the disease will appear in the next "x" months/years) and the other would include diagnostic tests (if the disease is present). The objectives and planning of your analysis should take into consideration this differentiation.

Thank you for this comment. We agree that methylation markers have shown good potential for use as predictive markers. However, for the purpose of this study, we plan to analyse methylation as diagnostic markers only. The reason for this is that we want to analyse methylation in the context of cervical screening, where a diagnosis of CIN2 or worse would equate to the necessity for local surgical cervical treatment.

In this context in our manuscript, when we refer to prospective studies we are referring to studies where methylation testing has been taken in conjunction with and prior to a cytological or histological diagnosis, and retrospective studies where methylation markers have been applied in the context of a known cytological or histological diagnosis. An analysis of predictive potential of methylation markers is beyond the scope of this diagnostic test accuracy review, therefore we do not refer to prospective studies in the context of predictive markers i.e. the risk of developing CIN2 or worse in the future.

We have added clarification in this section in the manuscript (page 8, line 23):

'Randomised control trials, cross-sectional studies, longitudinal cohort studies, and both prospective and retrospective case-control studies, will be included where they assess the diagnostic test accuracy of human or HPV methylation marker to detect high-grade pre-invasive cervical disease as confirmed by current histology as the gold standard. We will include studies assessing the accuracy of methylation markers for diagnosis only; in this review we will not include studies assessing methylation markers as predictive markers for future development of cervical cancer or pre-invasive cervical disease.'

8. In Page 10 / "Data extraction": there is no mention about including the version of genome assembly (e.g. hg38) for the CpG locations and the strategy to convert everything to a common assembly (e.g. LiftOver)

Thank you for this comment. We have added a sentence to the data extraction sentence to clarify this point (page 10, line 25):

'We will use hg38, and where studies have used hg19 or hs1, we will use LiftOver to convert these to hg38, using a LiftOver package in R.'

9. In Page 11 / lines 49-51: "We will assess... large studies only". I think this sentence needs clarification. Do you mean you will check whether only small studies can be useful or only large studies can be useful?

Thank you for this comment. We plan on conducting a sensitivity analysis to assess whether small studies have possibly over-emphasised our results; we want to assess whether the results would be similar if only large studies were included. In order to do this it will be necessary to use an arbitrary cut-off. We have updated the manuscript to make this clearer as follows (page 11, line 15):

'We will assess whether only small studies (those with less than 100 participants) provide supporting evidence and will conduct a sensitivity analysis to assess robustness of results using large studies only (studies with 100 participants or more).'

10. In Page 17 / lines 12-13: There seem to be a problem with the references' format

Thank you for your detailed review. We have amended this.

11. In Page 19 / Appendix 1: The search strategy needs some clarification. It seems that some lines are mixed up.

Thank you for your detailed review. We have updated the search terms slightly as Cervical Intraepithelial Neoplasia is no longer a subject heading on Medline in 2023. We have updated this Appendix 1. We have checked the order of search terms including with our college librarian and do not believe there is any further error but if the reviewer were able to provide more detail we could investigate this further.

Reviewer: 1

Competing interests of Reviewer: Minority shareholder of Sola Diagnostics GmbH, which holds a license to the intellectual property that protects the commercialization of the WID-CIN/WID-qCIN tests.

Reviewer: 2

Competing interests of Reviewer: I declare no competing interests.

VERSION 2 – REVIEW

REVIEWER	Herzog, Chiara University of Innsbruck, European Translational Oncology Prevention and Screening Institute
REVIEW RETURNED	12-May-2023

GENERAL COMMENTS	Dear authors, thank you for addressing and clarifying all points raised in the previous round of revision. The manuscript is substantially clarified and I recommend acceptance for publication.
---

REVIEWER	Ladoukakis , Efthymios Queen Mary University of London, Wolfson Institute of Population Health
REVIEW RETURNED	13-May-2023

GENERAL COMMENTS

This a well written strategy for a future meta-analysis study for evaluating the accuracy of current DNA methylation markers to diagnose CIN2+ abnormalities and cervical cancer. The authors have answered promptly all of my concerns and ammended their manuscript accordingly. Therefore, I propose the publication of their research protocol and I eagerly await its immediate application as well as the subsequent results.